# Uridine Diphosphate Glucuronosyl Transferase 2B28 (UGT2B28) Promotes Tumor Progression and Is Elevated in African American Prostate Cancer Patients

**DOI:** 10.3390/cells11152329

**Published:** 2022-07-29

**Authors:** Anindita Ravindran, Kimiko L. Krieger, Akash K. Kaushik, Hélène Hovington, Sadia Mehdi, Danthasinghe Waduge Badrajee Piyarathna, Vasanta Putluri, Paul Basil, Uttam Rasaily, Franklin Gu, Truong Dang, Jong Min Choi, Rajni Sonavane, Sung Yun Jung, Lisha Wang, Rohit Mehra, Nancy L. Weigel, Nagireddy Putluri, David R. Rowley, Ganesh S. Palapattu, Chantal Guillemette, Louis Lacombe, Éric Lévesque, Arun Sreekumar

**Affiliations:** 1Department of Molecular and Cell Biology, Baylor College of Medicine, 120D, Jewish Building, Houston, TX 77030, USA; anindita.hp@gmail.com (A.R.); kimiko.krieger@bcm.edu (K.L.K.); nitrr.akash@gmail.com (A.K.K.); danthasinghewaduge.piyarathna@bcm.edu (D.W.B.P.); basil.paul@bcm.edu (P.B.); uttam.rasaily@bcm.edu (U.R.); franklin.gu.13@gmail.com (F.G.); tdang@bcm.edu (T.D.); rajni.sonavane@gmail.com (R.S.); nweigel@bcm.edu (N.L.W.); putluri@bcm.edu (N.P.); drowley@bcm.edu (D.R.R.); 2Faculty of Medicine, Centre de Recherche du Centre Hospitalier Universitaire de Québec-Université Laval Research Center (CRCHUQc-UL) and Université Laval, Québec, QC G1V 4G2, Canada; helene.hovington@crchudequebec.ulaval.ca (H.H.); sadia.mehdi@crchudequebec.ulaval.ca (S.M.); louis.lacombe@crchudequebec.ulaval.ca (L.L.); eric.levesque@crchudequebec.ulaval.ca (É.L.); 3Advanced Technology Core, Baylor College of Medicine, Houston, TX 77030, USA; vputluri@bcm.edu; 4Verna and Marrs McLean Department of Biochemistry and Molecular Biology, Baylor College of Medicine, Houston, TX 77030, USA; jongmin.choi@bcm.edu (J.M.C.); syjung@bcm.edu (S.Y.J.); 5Michigan Center for Translational Pathology, Ann Arbor, MI 48109, USA; lishawang9@gmail.com (L.W.); mrohit@med.umich.edu (R.M.); 6Rogel Cancer Center, Michigan Medicine, Ann Arbor, MI 48109, USA; gpalapat@med.umich.edu; 7Department of Pathology, University of Michigan Medical School, Ann Arbor, MI 48109, USA; 8Center for Translational Metabolism and Health Disparities (C-TMH), Baylor College of Medicine, Houston, TX 77030, USA; 9Dan L. Duncan Comprehensive Cancer Center, Baylor College of Medicine, Houston, TX 77030, USA; 10Department of Urology, University of Michigan Medical School, Ann Arbor, MI 48109, USA; 11Faculty of Pharmacy, Pharmacogenomics Laboratory, Centre de Recherche du Centre Hospitalier Universitaire de Québec-Université Laval Research Center (CRCHUQc-UL) and Université Laval, Québec, QC G1V 4G2, Canada; chantal.guillemette@crchudequebec.ulaval.ca

**Keywords:** prostate cancer, glucuronidation, UGT2B28, African American prostate cancer, androgen signaling, metabolic regulation, tumorigenesis

## Abstract

Prostate cancer (PCa) is the second most diagnosed cancer in the United States and is associated with metabolic reprogramming and significant disparities in clinical outcomes among African American (AA) men. While the cause is likely multi-factorial, the precise reasons for this are unknown. Here, we identified a higher expression of the metabolic enzyme UGT2B28 in localized PCa and metastatic disease compared to benign adjacent tissue, in AA PCa compared to benign adjacent tissue, and in AA PCa compared to European American (EA) PCa. *UGT2B28* was found to be regulated by both full-length androgen receptor (AR) and its splice variant, AR-v7. Genetic knockdown of UGT2B28 across multiple PCa cell lines (LNCaP, LAPC-4, and VCaP), both in androgen-replete and androgen-depleted states resulted in impaired 3D organoid formation and a significant delay in tumor take and growth rate of xenograft tumors, all of which were rescued by re-expression of UGT2B28. Taken together, our findings demonstrate a key role for the *UGT2B28* gene in promoting prostate tumor growth.

## 1. Introduction

Prostate cancer (PCa) is the leading cause of cancer incidence in men in the United States. PCa is associated with disparities in patient outcomes wherein African American (AA) men have an earlier onset and poorer clinical outcomes compared to European American (EA) men [1]. PCa growth is dependent on ligand-associated AR signaling that renders them susceptible to therapies that target androgen signaling, such as androgen-deprivation therapy (ADT), which are widely used in the clinic to treat patients as first-line treatment for metastatic disease. Over time, however, these tumors develop resistance to ADT, and invariably develop to a state known as castration resistant prostate cancer (CRPC). The development of CRPC is thought to occur via a variety of mechanisms that may involve ligand-independent AR signaling inclusive of pathways that utilize AR variants that lack the ligand binding domain (e.g., AR-v7). Our group previously demonstrated metabolic reprogramming as a key component of PCa growth and metastatic progression [2,3,4,5,6]. Further, using a combination of targeted mass spectrometry and metabolic phenotyping microarrays, our laboratory previously found that the uridine diphosphate glucuronosyltransferase (UGT) pathway is one of the major altered metabolic pathways in both hormone-sensitive and castration-resistant prostate cancer [4]. Moreover, in the same study, we found UGT2B28 expression to be associated with PCa progression [4], which characterized its tumor promoting role.

The UGT family of enzymes is responsible for androgen glucuronidation, which regulates the steady-state levels of androgens in the body [7]. Decreased UGT2B15 expression and increased UGT2B17 expression were shown to promote PCa progression [8]. UGT2B28 is a key member of this group of UGTs whose mechanistic role in prostate cancer remains uncharacterized. In this study, we described the clinical relevance of UGT2B28, compared its expression between AA and EA PCa, identified a regulatory role for androgen signaling in UGT2B28 expression, demonstrated a tumor promoting function for the gene independent of the presence of androgens, and provided insights into potential effector pathways associated with its tumor promoting function. 

## 2. Materials and Methods

### 2.1. Reagents

The UGT2B28 antibody for the tissue microarray analysis shown in Figure 1A was purchased from Abnova (Cat. #H00054490-B01P). The UGT2B28 antibody (Ab2321) used for tissue microarray data in Figure 1B–D and Figure 2D,E was custom made by Dr. Levesque and Guillemette according to the previously published method [9]. AR and PSA antibodies were purchased from Santa Cruz Biotechnology. Antibodies for beta-actin and GAPDH were purchased from Sigma Aldrich.

### 2.2. Cell Lines

LNCaP cells were obtained from the Tissue Culture Core at Baylor College of Medicine. LAPC-4 cells were a gift from Dr. Daniel Frigo (MD Anderson) and VCaP cells were a gift from Dr. Sean McGuire (Baylor College of Medicine). All cell lines were short tandem repeat (STR)-types from the MD Anderson Cytogenetics and Cell Authentication Core and were regularly confirmed to be free of mycoplasma contamination using the MycoAlert ™ Mycoplasma Detection Kit (Lonza, Anaheim, CA, USA). 

### 2.3. Cell Culture

LNCaP cells were grown in RPMI-1640 media (Invitrogen Corp., Carlsbad, CA, USA) supplemented with 10% fetal bovine serum (FBS, Hyclone Labs, Thermo Scientific, Rockford, IL, USA) and 1% penicillin-streptomycin (Hyclone Labs, Thermo Scientific, Rockford, IL, USA). LAPC-4 cells were grown in IMDM (Invitrogen Corp., Carlsbad, CA, USA) supplemented with 15% FBS, 1% penicillin-streptomycin, and 1 nM R1881 (Sigma Aldrich, St. Louis, MO, USA). VCaP cells were grown in DMEM (Invitrogen Corp., Carlsbad, CA, USA) and supplemented with 10% FBS and 1% penicillin-streptomycin. All cells were maintained at 37 °C, 5% CO_2_, and 95% humidity.

### 2.4. Lentiviral Transduction

To generate a stable knockdown of UGT2B28 in LNCaP, LAPC-4, and VCaP cells, cells were transduced with two independent clones of lentiviral UGT2B28 shRNA (Clone ID: V3LHS_378777) and non-targeting control pGIPZ shRNA purchased from the Cell-Based Assay Screening Service Core (Baylor College of Medicine). Transduction was carried out at an MOI of 5, and cells containing the KD were stably selected with 1 µg/mL (LNCaP, LAPC-4), and 2 µg/mL (VCaP) puromycin from Sigma Aldrich (Cat. #P8833) in their respective media. Similarly, stable overexpression of UGT2B28 in NT and KD cells (NTOE, KD2OE, and KD4OE) was achieved using over-expression lentiviral vector for UGT2B28 (LPP-H0607-Lv157-050) along with a negative control (LPP-NEG-Lv157-050) purchased from Genecopoeia at an MOI of 5 and the overexpressing cells were stably selected using neomycin from Sigma Aldrich (1 mg/mL). 

### 2.5. Organoid Forming Assay

LNCaP and LAPC-4 cells were seeded at a density of 600,000 cells in 300 µL media on 12 mm Millicell™ inserts (Millipore Sigma, St. Louis, MO, USA) with cellulose ester filters. The inserts were placed in 24-well plates with each well containing 600 µL media at the bottom and the cells were allowed to spontaneously form organoids by 16 h (LNCaP) and 24 h (LAPC-4). The organoids were imaged using a fluorescence confocal microscope with brightfield and GFP filters at 4× and 10× magnifications at the Integrated Microscopy Core at Baylor College of Medicine. 

### 2.6. Perturbation of Androgen Signaling in the Organoid Assay

LNCaP cells were seeded and grown in RPMI-1640 media as described above in 6-well plates for 24 h on Day 1. The medium was changed to RPMI-1640 media (Invitrogen Corp., Carlsbad, CA, USA) lacking phenol red and supplemented with 10% charcoal-stripped FBS (CSS, Invitrogen Corp. Carlsbad, CA, USA) and 1% penicillin-streptomycin on Day 2 for another 24 h. On Day 3, cells were treated with CSS, 1 nM 5α-dihydrotestosterone (DHT) (Sigma Aldrich, St. Louis, MO, USA), 10 μM enzalutamide (MDV-3100) (Sigma Aldrich, St. Louis, MO), 300 nM ARCC4 (Sigma Aldrich, St. Louis, MO, USA), 1 nM DHT + 10 μM MDV-3100, or FBS for 24 h. On Day 4, the treated cells were trypsinized and reseeded onto Millicell ™ inserts in their respective media and were allowed to form organoids for another 24 h. The organoids were imaged as described above.

### 2.7. Immunohistochemistry Analysis

#### 2.7.1. University of Michigan Prostate Cancer Tissue Microarrays

Established clinically annotated tissue microarrays (TMAs) created from tissues obtained during radical prostatectomy (*n* = 165; benign, *n* = 95 and localized PCa, *n* = 70) with progression endpoints, such as biochemical recurrence, were used for immunohistochemistry analysis. TMAs composed of metastatic samples (CRPC) that were also used to measure UGT2B28 protein expression by immunohistochemistry were obtained from the University of Michigan’s warm autopsy program which includes 45 CRPC tissues sampled across multiple sites, with each site replicated into 3 cores (i.e., prostate, liver, lymph node, soft tissue, dura, bladder, adrenal gland, seminal vesicle, diaphragm, and pancreas). UGT2B28 antibody for the staining of these TMAs was purchased from Abnova. Immunohistochemistry staining was performed at the histopathology core at University of Michigan. Immunostaining was categorized into weak, moderate, and strong staining by genitourinary pathologists Dr. Rohit Mehra and Dr. Lisha Wang.

#### 2.7.2. Baylor College of Medicine Prostate Cancer Tissue Microarrays

An analysis of UGT2B28 expression in tissue microarrays containing AA and EA PCa patients (*n* = 105 AA patients, *n* = 102 EA patients) was conducted with a custom made UGT2B28 antibody as previously described [9]. Matched tumor-benign pairs of prostate tissues from patients were processed into formalin-fixed paraffin-embedded tissues and organized into tissue microarrays. Custom made UGT2B28 antibody (Ab2321, 1:500 dilution) was used to stain tissue microarrays with Autostainer Link 48 from DAKO using the EnVision FLEX10 protocol (Agilent, Santa Cara, CA, USA). The intensity of staining was scored for both nuclear and cytoplasmic compartments. Both the percentages of negative and positive nuclei were recorded, as well as the percentage of stained cytoplasm.

### 2.8. Immunofluorescence

Immunofluorescence experiments for UGT2B28 were performed using custom made UGT2B28 antibody. Briefly, cells were plated on poly-d-lysine coated slides then fixed in 4% paraformaldehyde and permeabilized with 1× PBS–0.2% Triton X-100. After blocking with 1× PBS-5%-BSA-5% goat serum, cells were incubated with anti-UGT2B28 (1:200) and anti-Androgen Receptor antibody (1:200, 441 from Santa Cruz, Dallas, TX, USA) and incubated with secondary antibodies, including Alexa Fluor 488-labeled goat anti-rabbit antibody and Alexa Fluor 555-labeled goat anti-mouse antibody (1:1000) (Invitrogen, Burlington, ON, Canada). Cells were stained with DRAQ5 (1:2000) (Abcam, Branford, CT, USA). Images were captured using a LSM510 META NLO laser scanning confocal microscope (Zeiss, Toronto, ON, Canada). Zen 2009 software version 5.5 SP1 (Zeiss, Toronto, ON, Canada) was used for image acquisition.

### 2.9. Mouse Xenograft Studies

Animal well-being and animal experiments were performed and monitored in accordance with a specific animal protocol approved by the Institutional Animal Care and Use Committee (IACUC) of Baylor College of Medicine. Six-week-old athymic nude mice (Strain #553), both intact and castrated, were purchased from Charles River, Frederick, MD. The 60-day release testosterone pellets for immuno-deficient mice (Cat. #SA-151) were purchased from Innovative Research of America, Sarasota, FL. Mice were subcutaneously implanted with pellets (12.5 mg/kg body weight) and administered subcutaneous injections of a mixture of 2,000,000 tumor cells (LNCaP, LAPC-4, or VCaP), 50,000 HPS-19I prostate stromal cells [10], and Matrigel (Corning, Tewksbury, MA, USA). Tumors were measured three times a week. LNCaP tumors were resected at ~500 mm^3^ to observe KD growth and LAPC-4 and VCaP tumors were resected when the control tumors reached ~600 mm^3^ to obtain relative tumor take curves. Similar subcutaneous injections were performed in castrated mice for VCaP tumors. 

### 2.10. Quantitative Real-Time Polymerase Chain Reaction

RNA extraction from cells or tissues was performed using the RNeasy Mini Kit from QIAGEN. RNA was reverse transcribed into cDNA using the cDNA Superscript Mix (Quanta Biosciences, Cat. #95048-500) and RT-qPCR was performed using SYBR green (Life Technologies, Cat. #4385614). Either 18S or β-actin were used as appropriate house-keeping controls. The primers used in this study are listed in Appendix A.

### 2.11. Sample Preparation for Mass Spectrometry

LNCaP cells were starved in RPMI without glucose (Invitrogen Corp., Carlsbad, CA, USA) for 24 h and treated with 12 mM ^13^C-glucose (Sigma Aldrich, St. Louis, MO, USA) containing RPMI media for an additional 24 h. Cells were washed three times with ice-cold PBS and pelleted. Frozen pellets were thawed on ice and subjected to lysis by repeated cycles of freeze–thaw in liquid nitrogen and at room temperature. A total of 750 µL of ice-cold methanol:water (4:1) with 20 µL of spiked internal standard was added to each sample and homogenized for 30 s pulses of 1 min each. A total of 450 µL of ice-cold chloroform and 150 µL of water were sequentially added to each sample. Both the organic and aqueous layers were combined and filtered via a 3 kDa Amicon Ultracel molecular filter (Millipore, Billerica, MA, USA). The filtrates were then vacuum-dried using a Genevac EZ-2 plus (Gardinier, NY, USA) and resuspended in 100 µL of 1:1 methanol:water with 0.1% formic acid before injecting the samples into the mass spectrometer. 

### 2.12. Flux Measurement and Analysis of UGT Co-Substrates

The method for the flux measurement of intracellular UGT metabolites was generated using uridine 5′-diphosphoglucuronic acid as our reference standard (Cat. #U6751-100MG), and the transitions that were monitored are tabulated in Appendix A.

### 2.13. Statistical Analysis

Unless otherwise stated, all samples were assayed in triplicate. All in vitro experiments were repeated for a minimum of three independent runs. Unless otherwise indicated, data are represented as mean ± standard deviation (SD), and significance was calculated using Student’s unpaired two-tailed *t*-test.

## 3. Results

### 3.1. UGT2B28 Expression Is Associated with Advanced Disease and Elevated in AA PCa

Metabolic profiling of androgen dependent and independent cell lines performed in our laboratory previously revealed two important metabolic pathways that were rewired in PCa—the amino sugar pathway (hexosamine biosynthetic pathway or HBP) and the UGT pathway [4]. Our lab has since described a role for reduced amino sugar via the hexosamine biosynthetic pathway in CRPC progression [11]. The UGT pathway was the second most significantly altered pathway in PCa cells that we observed [4]. Multiple UGT family members were identified, including UGT2B28, a member that has been reported to be associated with early biochemical recurrence but not characterized to date for its tumor promoting function [9]. Immunohistochemical staining of UGT2B28 protein expression in a TMA containing tissues from benign prostate, localized tumors, and metastatic tumors, revealed an increase in UGT2B28 expression in localized tumors vs. benign adjacent prostate tissue and metastatic vs. localized tumors (Figure 1A). Furthermore, we also found that AA PCa patients had significantly higher levels of nuclear UGT2B28 expression than EA PCa patients (Figure 1B, Appendix A), and a significantly elevated expression in the cytoplasmic compartment for the same comparison (Figure 1C). Within AA PCa, tumors had significantly elevated expressions of UGT2B28, both in the nucleus and cytoplasm, compared to benign adjacent prostate tissue (Figure 1D). Interestingly, there was no difference in the nuclear and cytoplasmic expression of UGT2B28 for AA vs. EA benign adjacent tissues (Appendix A), suggesting tumor-specific alterations of UGT2B28 in the AA vs. EA comparison. 

**Figure 1 cells-11-02329-f001:**
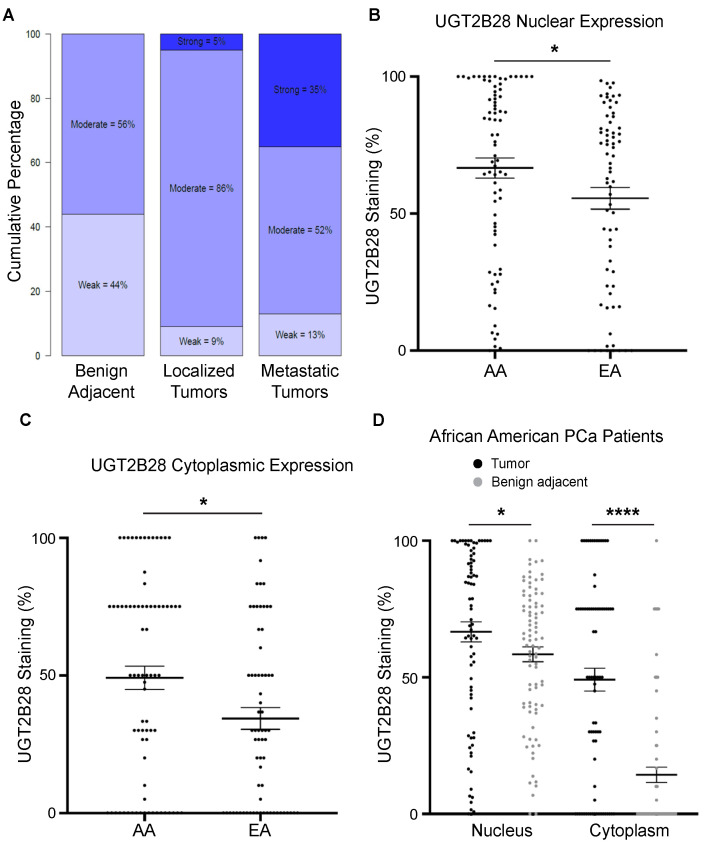
Increased UGT2B28 protein expression during prostate cancer progression and in African American prostate cancer patients. (**A**). Percentage of overall staining of UGT2B28 in benign adjacent tissue, localized tumors, and metastatic tumors. (**B**). Percentage of staining of nuclear UGT2B28 in African American vs. European American prostate tumors. * *p* < 0.05. (**C**). Percent staining of cytoplasmic UGT2B28 in African American vs. European American prostate tumors. * *p* < 0.05. (**D**). Percentage of staining of nuclear and cytoplasmic UGT2B28 in African American benign adjacent prostate tissue and prostate tumors. * *p* < 0.05, **** *p* < 0.0001. The data are represented as the mean +/− the standard error of the mean (SEM).

### 3.2. Androgen Receptor Regulates UGT2B28 Expression in PCa

To characterize the role of UGT2B28 in PCa, we established a series of genetic models in LNCaP, LAPC-4, and VCaP cells, in which we knocked down (KD) UGT2B28 expression using lentiviral shRNA (Appendix A) and rescued the expression by means of lentiviral-based re-expression of UGT2B28 (UGT2B28 R) (Figure 2A–C). Given the high degree of sequence overlap between members of the UGT2B family, we used a qPCR approach to confirm that our re-expression rescue construct was specific only to UGT2B28. As shown in Appendix A, the re-expression of UGT2B28 rescued the mRNA levels of UGT2B28 but not UGT2B10, UGT2B11, UGT2B15, and UGT2B17. 

**Figure 2 cells-11-02329-f002:**
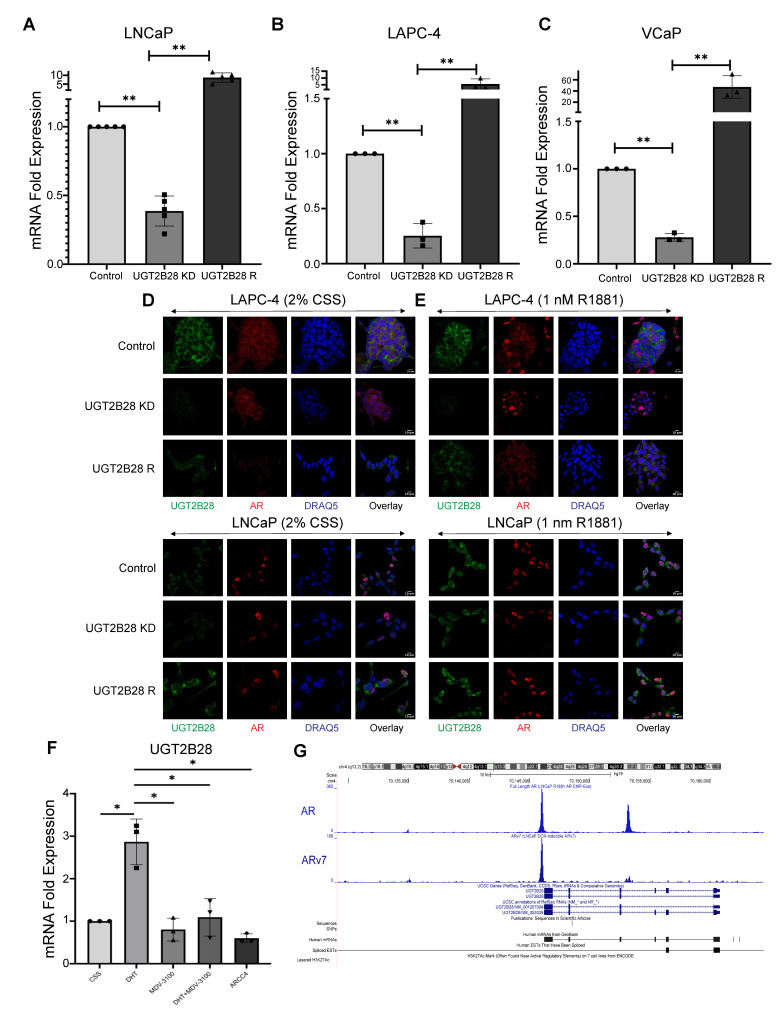
(**A**–**C**). qPCR verification of UGT2B28 transcript levels in (**A**) LNCaP, (**B**) LAPC-4 and (**C**) VCaP cells upon shRNA-mediated UGT2B28 knockdown (UGT2B28 KD) or re-expression rescue (UGT2B28 R), compared to scrambled controls (UGT2B28 NT, ** *p* < 0.001, *n* = 3/group/cell line). The data are represented as the mean +/− the standard deviation (SD). (**D**). Immunofluorescence images showing UGT2B28 and AR expression and their localization in LNCaP and LAPC-4 cells containing control (UGT2B28 NT), UGT2B28 KD, and re-expression rescue (UGT2B28 R) grown in charcoal-stripped serum (CSS). Scale bar = 10 μm. (**E**). Same as in (**D**), but for LNCaP and LAPC-4 cells grown in 1 nM R1881. Scale bar = 10 μm. (**F**). UGT2B28 transcript expression levels in LNCaP cells treated with 1 nM 5α-dihydrotestosterone (DHT), 10 µM MDV-3100, 1 nM DHT+10 µM MDV-3100, and 300 nM ARCC4 for 24 h. The data are represented as the mean +/− the standard deviation (SD). * *p* < 0.01, *n* = 3. (**G**). ChIP-Exo of the UGT2B28 promoter region in LNCaP cells treated with 1 nM R1881 (full length AR binding, (top)) and LNCaP cells with doxycycline-induced AR-v7 (AR-v7 binding, (bottom)) reveal binding of AR and AR-v7, respectively.

We next examined the co-substrate levels of UGT2B28 in the LNCaP model. The UGT pathway is a branch pathway of glycolysis that involves the transfer of carbon skeleton from glucose to glucose-1-phosphate, which is then transferred to UDP-glucose. The latter is then converted to UDP-glucuronic acid, which forms the co-factor for glucuronidation reactions catalyzed by UGT members. We labeled LNCaP cells with [U]-^13^C-Glucose for 24 h and measured the percentage of incorporation of ^13^C label in UDP-glucuronic acid relative to the total pool. As expected, our results confirmed a significant increase in the percentage of ^13^C-labeled UDP-glucuronic acid in KD cells (due to metabolite accumulation) that was then rescued to control levels upon the re-expression of the protein (Appendix A). This further confirmed that our genetic models demonstrated expected changes in UGT co-substrate concentrations as a surrogate of enzymatic function. 

To confirm UGT2B28 KD and rescue at the protein level, we used immunofluorescence to test our genetic models. We used LNCaP and LAPC-4 cells grown in charcoal-stripped serum (CSS). UGT2B28 expression was reduced in UGT2B28 KD cells and rescued in the UGT2B28 R cells (Figure 2D). Furthermore, in both models, UGT2B28 was localized to both nuclear and cytoplasmic compartments (Figure 2D). 

To evaluate the role of androgen signaling on UGT2B28 expression and localization further, we first used immunofluorescence staining to determine the expression of UGT2B28 in the LNCaP and LAPC-4 genetic models in the presence of synthetic androgen R1881. UGT2B28 expression was increased and localized to the nucleus and cytoplasm in both LNCaP and LAPC-4 genetic models upon R1881 treatment (Figure 2E). To further test the role of androgens in regulating UGT2B28 expression, we used LNCaP cells grown in CSS, 5α-dihydrotestosterone (DHT), enzalutamide (MDV-3100), DHT in combination with enzalutamide (DHT + MDV-3100), and ARCC4, a PROTAC specific for AR degradation. Using qPCR, we observed that UGT2B28 mRNA expression was elevated upon DHT treatment (Figure 2F). UGT2B28 expression, however, was significantly reduced to baseline levels observed in CSS media upon treatment with either MDV-3100 alone, DHT + MDV-3100, or ARCC4 (Figure 2F). These results were similar to the patterns observed in androgen-responsive PSA mRNA expression under the same treatment conditions (Appendix A), suggesting that UGT2B28 expression was regulated by ligand-dependent AR signaling. This was further confirmed by the in silico analysis of chromatin immunoprecipitation sequencing (ChIP-Exo) studies for both full length AR and AR-v7 made publicly available by Dr. Paul Basil. Our analysis revealed that in LNCaP cells treated with 1 nM R1881, full length-AR directly binds to the UGT2B28 promoter (Figure 2G). In LNCaP cells containing doxycycline-inducible AR-v7 expression, our analysis of the data showed that AR-v7 also strongly binds the UGT2B28 promoter (Figure 2G). These results suggest that UGT2B28 is directly regulated by both full length-AR and AR-v7 signaling in PCa cells.

### 3.3. UGT2B28 Deficiency Impairs Organoid Formation in AD Cells In Vitro

Having established genetic models, we then studied phenotypic changes induced by the KD of UGT2B28 that were rescued by the re-expression of the protein. We sought to examine the ability of the cells to form 3D organoids using low attachment plates. We used Millicell inserts with cellulose ester filters to establish a low attachment setting for cells to form organoids. LNCaP and LAPC-4 cells transduced with non-targeting (NT) control vector spontaneously formed organoids within 16–24 h post-seeding (Figure 3A,B). In contrast, UGT2B28 KD cells failed to form such organoids. Specifically, the organoids formed by the UGT2B28 KD cells exhibited poor cell–cell adhesion, were loosely packed, and continued to remain attached to the edges of the Millicell insert (Figure 3A,B, Appendix A). However, the re-expression of UGT2B28 in the KD setting (UGT2B28 R) rescued the above phenotype and formed organoids that phenocopied those formed by the control cells (NT) (*n* = 3). These findings implicated a role for UGT2B28 in cell–matrix interactions.

### 3.4. UGT2B28 Deficiency Prolongs Tumor Take and Delays Tumor Growth In Vivo

Next, we examined the role of UGT2B28 on tumor growth in vivo. We subcutaneously injected control (NT)-, UGT2B28 KD-, and UGT2B28 R-transduced LNCaP, LAPC-4 and VCaP cells in athymic nude mice pre-implanted with 12.5 mg/kg testosterone pellets (Figure 3C–E). The cells were injected along with Matrigel and HPS-19I prostate stromal cells [10]. Mice injected with control cells were the first to form tumors in all xenograft models. Specifically, these control cells formed palpable tumors in 7, 5, and 20 days post-injection in the LNCaP, LAPC-4 and VCaP xenograft models, respectively. In contrast, the UGT2B28 KD cells across the three models exhibited a significant delay (an additional 2–11 days) in tumor take rates, with the palpable tumors appearing on Days 10, 14, and 32 post-injection in the LNCaP, LAPC-4 and VCaP xenograft models, respectively. Additionally, the KD tumors grew at a significantly slower rate compared to the control (NT) tumors in all three models. Importantly, both the tumor take and tumor growth rate were completely rescued in all three xenograft models upon the re-expression of UGT2B28 (UGT2B28 R, Figure 3C–E). The qPCR analysis of the tumors confirmed the expected expression patterns for UGT2B28 in the different genetic models (Figure 3C–E, refer to the inset). These findings suggest a critical role for UGT2B28 in PCa tumor take and tumor growth.

### 3.5. Effects of UGT2B28 on Organoid Formation and Tumor Growth Is Not Dependent on Ligand-Driven AR Signaling

Given the regulation of UGT2B28 expression by AR, we sought to determine whether the organoid formation and tumor growth phenotypes that we observed in AD cell lines in response to UGT2B28 manipulation were dependent on androgen-bound AR signaling. To test this, we treated LNCaP NT, UGT2B28 KD, and UGT2B28 R cells grown in CSS media with either 1 nM DHT, 10 μM MDV-3100, 1 nM DHT+10 μM MDV-3100, or FBS for 24 h, and allowed them to form organoids for an additional 24 h (*n* = 3). We found that treatment with DHT promoted the moderate proliferation of cells in organoids formed by all three genetic models. However, enhancing or attenuating AR signaling by modulating androgen levels resulted in organoids that phenocopied our results shown in Figure 3A,B (Figure 4A). Furthermore, the organoid formation remained unchanged compared to our findings in Figure 3A,B, even in the presence of 300 nM ARCC4 for 24 h, further confirming that the downstream phenotypic effect of UGT2B28 on organoid formation was independent of canonical, ligand-dependent AR signaling (Figure 4A). 

To test whether the UGT2B28-associated tumor growth was reliant on AR signaling, we subcutaneously injected VCaP NT, UGT2B28 KD, and UGT2B28 R cells in castrated mice (Figure 4B). Interestingly, even though castration delayed the tumor take rate across the groups by 23–33 days, the overall pattern of tumor growth was similar to the results obtained in mice implanted with androgen pellets (refer Figure 3C–E). Specifically, even in the castrated setting, the KD tumors had a significantly delayed tumor take and reduced tumor growth rate, compared to control and UGT2B28 R tumors (Figure 4B). Taken together, our findings demonstrated a key role for UGT2B28 in regulating 3D organoid formation in vitro and xenograft tumors in vivo across multiple PCa models. 

## 4. Discussion

Metabolic reprogramming has been established as a hallmark of cancer and has become a new focus in understanding the molecular underpinnings that drive tumorigenesis [12]. The goal of our laboratory was to define these dysregulated metabolic pathways that contribute to cancer cell growth and can be exploited for possible cancer therapeutics. Our laboratory delineated the altered metabolites and associated pathway changes associated with AD and CRPC cell lines, among which the UGT pathway emerged as one of the most significant [4]. UGT2B28, a previously uncharacterized UGT family member, was among the chief altered enzymes associated with this pathway in our profiling results. We demonstrated that UGT2B28 expression is strongly correlated with disease aggressiveness, as previously reported [9]. Furthermore, we identified elevated levels of UGT2B28 in AA PCa patients compared to their EA PCa counterparts, with the former known to have worse PCa-associated clinical outcomes [1].

It is particularly interesting that our results show that UGT2B28 is regulated by AR and AR-v7 at the genetic level, and its tumor promoting role is not fully compromised by castrated mouse models or the ablation of AR or androgens. We found that MYC is a remarkable example of another AR target gene with a tumorigenic function in PCa that is independent of androgen stimulation [13]. 

UGT2B28 has been described to promote PCa progression and was linked with alteration in circulating androgen levels in men with localized PCa [9]. However, our in vitro and in vivo functional data revealed similar effects on 3D organoid formation and a reduction in tumor growth upon UGT2B28 KD, both in a ligand-driven and castrated AR signaling environment, which suggests that UGT2B28 may have non-canonical roles in prostate tumorigenesis. An insight into this non-canonical function was obtained by examining the flux measurements of ^13^C-UDP-glucuronic acid (Appendix A). We previously determined that UDP-glucuronic acid, the co-substrate of UGT2B28, is a precursor for hyaluronic acid and the depletion of this precursor reduces tumor growth [14]. Hyaluronan synthase 2 (HAS2) is the enzyme responsible for synthesizing hyaluronic acid, and hyaluronidase-1 (HYAL1) is the enzyme that cleaves high molecular weight hyaluronic acid, which is anti-tumorigenic, into smaller molecular weight tumor promoting fragments known to promote angiogenic, tumorigenic, and migratory phenotypes in many different cancers [15,16]. It is possible that an altered balance between HAS2 and HYAL1 upon UGT2B28 KD may contribute to the delay in tumor take and tumor growth, which would need to be verified in the future using additional experiments. We investigated signaling pathways that may be altered upon UGT2B28 expression using mass spectrometry-based proteomics experiments conducted on both VCaP tumors (in intact mice) and LNCaP cells (Appendix A).

One of the challenges in studying UGT2B family members for their role in PCa is their extensive sequence overlap. This has led to a current lack of commercial mono-specific antibodies to UGT2B28. We, along with our collaborators, addressed this by synthesizing a custom-made UGT2B28 antibody and we performed mass spectrometry experiments to further confirm the specificity of our UGT2B28 rescue. We also demonstrated that the UGT2B28 R setting results in the rescue of only UGT2B28 and none of the other family members. Furthermore, in all our studies, we mandated that the interpretation of the results was dictated by the ability of UGT2B28 R model to completely reverse the phenotype resulting from UGT2B28 KD. Finally, to ensure the robustness of our findings, we ensured that the phenotype of the 3D organoids and xenograft tumors resulting from the perturbation of UGT2B28 was identical in three independent PCa models. In light of all of the above, our data provide support for a non-canonical function of UGT2B28 in PCa. 

## 5. Conclusions

Our study highlighted the increased expression of UGT2B28 in AA PCa and described a non-canonical role for UGT2B28 in promoting PCa progression. 

## Figures and Tables

**Figure 3 cells-11-02329-f003:**
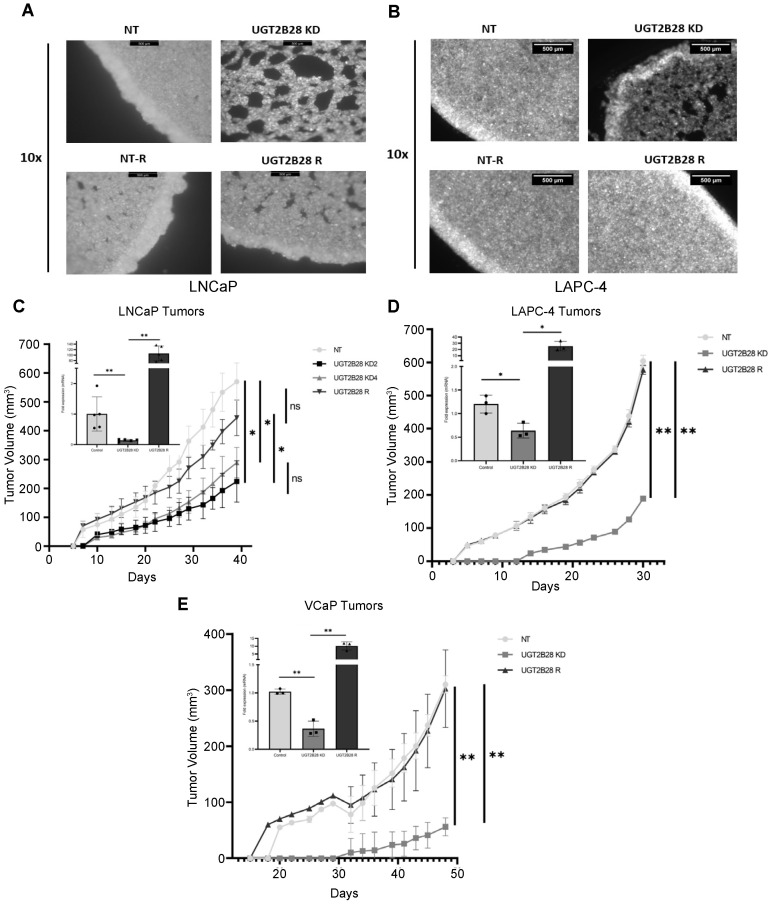
UGT2B28 deficiency in PCa cells impairs organoid formation in vitro and prolongs tumor take and growth in vivo. (**A**). 10× magnified Green Fluorescence Protein (GFP) images of 3D organoids formed from stably transduced LNCaP cells containing control (NT), UGT2B28 KD, UGT2B28 re-expression (Control + OE), and UGT2B28 re-expression rescue (UGT2B28 R) in the KD setting (UGT2B28 rescue). The 3D organoid formation was conducted in triplicates/group/cell line in Millicell inserts for 24 h. Scale bar = 500 μm. (**B**). Same as in (**A**), but for LAPC-4 cells. (**C**). Tumor growth curve of xenografts generated using LNCaP cells containing NT, UGT2B28 KD, and UGT2B28 R in athymic nude mice supplemented with testosterone. (**D**). Same as in (**C**), but for LAPC-4. (**E**). Same as in (**C**), but for VCaP. For panels (**C**–**E**), please refer to legends for number of replicates in each group. Inset shows verification of UGT2B28 transcript expression in xenograft tumors obtained from each group. The data are represented as the mean +/− the standard deviation (SD). * *p* < 0.01, ** *p* < 0.001, ns: not significant.

**Figure 4 cells-11-02329-f004:**
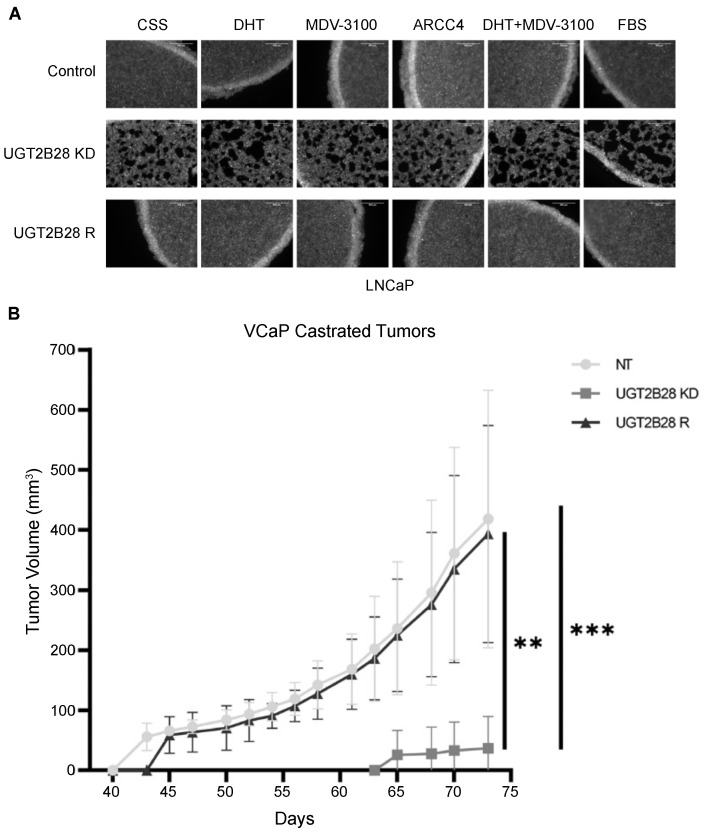
Prolonged tumor take and delayed tumor growth in response to UGT2B28 downregulation is independent of ligand dependent AR signaling. (**A**). 10× GFP images of LNCaP organoids formed from non-targeting (NT), UGT2B28 KD, and UGT2B28 overexpression in KD4 background (UGT2B28 R) cells in response to treatment with charcoal-stripped serum (CSS), 1 nM 5α-dihydrotestosterone (DHT), 10 μM enzalutamide (MDV-3100), 300 nM ARCC4, 1 nM DHT + 10 μM MDV-3100, and fetal bovine serum (FBS) for 24 h. *n* = 3. Scale bar = 500 μm. (**B**). Tumor growth curve for NT, UGT2B28 KD, and UGT2B28 R VCaP xenografts in castrated, athymic nude mice. The data are represented as the mean +/− the standard deviation (SD). ** *p* < 0.001, *** *p* < 0.0001. NT, *n* = 6, UGT2B28 KD, *n* = 7, UGT2B28 R, *n* = 6.

## Data Availability

The data presented in this study are available in this article (and Appendix A). ChIP-Exo data was previously uploaded to the GEO Database (GSE143906).

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
