# Peer review of "Uridine Diphosphate Glucuronosyl Transferase 2B28 (UGT2B28) Promotes Tumor Progression and Is Elevated in African American Prostate Cancer Patients"

_cells, 2022, doi:10.3390/cells11152329_

Round 1
Reviewer 1 Report
In this manuscript, Krieger et al have performed experiments to define the role of UGT2B28 in prostate cancer progression and its relationship with racial disparity in prostate cancer. The authors critically evaluated the role of UGT2B28 in vitro and in vivo using knockdown and over-expression models. Expression of UGT2B28 is increased in African American men when the cancer progresses from localized to metastatic disease. The data also suggest that UGT2B28 could be a tumor promoting protein and knocking it down will significantly reduce tumor growth.
The manuscript is well done with all controls and communicates an important conclusion that could potentially be translated to the clinic as either a diagnostic, prognostic, or a therapeutic marker. Few suggestions to improve the manuscript.
1. Could a correlation be derived between UGT2B28 expression and grade or stage or Gleason score or PSA from the respective clinical specimens?
The organoid culture images are not the best representation of the data. If the authors can quantify the images and provide the results in a graphical format, it will be helpful to interpret the data.
Author Response
Reviewer #1
Comments and Suggestions for Authors
In this manuscript, Krieger et al have performed experiments to define the role of UGT2B28 in prostate cancer progression and its relationship with racial disparity in prostate cancer. The authors critically evaluated the role of UGT2B28 in vitro and in vivo using knockdown and over-expression models. Expression of UGT2B28 is increased in African American men when the cancer progresses from localized to metastatic disease. The data also suggest that UGT2B28 could be a tumor promoting protein and knocking it down will significantly reduce tumor growth.
The manuscript is well done with all controls and communicates an important conclusion that could potentially be translated to the clinic as either a diagnostic, prognostic, or a therapeutic marker. Few suggestions to improve the manuscript.
- Could a correlation be derived between UGT2B28 expression and grade or stage or Gleason score or PSA from the respective clinical specimens?
The authors would like to thank the reviewer for their comments. PSA information from the respective clinical specimens is not available for both University of Michigan (Figure 1A) and Baylor College of Medicine cohorts (Figure 1B-1D). The clinical specimens from Baylor College of Medicine scored for this manuscript has Gleason grade information. Due to the qualitative scoring assessment for the University of Michigan samples, as well as a limited number of samples with matched Gleason information from the Baylor College of Medicine cohort, the authors could not conduct correlation analyses for all samples. The authors, however, were able to run a small-scale Spearman r correlation analysis for the Baylor College of Medicine samples that were quantitatively scored. There was no correlation observed between Gleason score and nuclear UGT2B28 expression or between Gleason score and cytoplasmic UGT2B28 expression within AA and/or EA patients. This could be due to the fact that most of the Baylor College of Medicine cohort had a composite Gleason score of 7 (since majority of the AA and EA tumors were Gleason matched), thus there would not be any correlations that could arise from comparisons with samples that would have similar Gleason scores.
- The organoid culture images are not the best representation of the data. If the authors can quantify the images and provide the results in a graphical format, it will be helpful to interpret the data.
The organoid images for UGT2B28 were used to display the phenotypic effects of organoid formation. This was a qualitative experiment that was to observe tightly packed cells—or lack thereof—within each organoid model genetically manipulated with UGT2B28 shRNA knockdown or UGT2B28 rescue. These effects may be also the reason why the authors observed decreased tumor take and tumor growth rates with the three mouse xenograft models. The authors regret to inform the reviewer that, due to the qualitative nature of the experimental design, quantification of the organoid images cannot be conducted for this experiment.
Reviewer 2 Report
In this study, the authors described UGT2B28 protein expression is associated with PCa progression and elevated in AA PCa patients compared with EA, and regulated by androgen receptor. They also investigated the role of UGT2B28 in organoid formation and tumor growth, and further demonstrated that the tumor promoting function of UGT2B28 is independent of ligand-driven AR signaling. The manuscript is scientifically sound, and the conclusions are consistent with the evidence presented. I only have some minor comments.
1. The authors presented the increased UGT2B28 protein expression during PCa progression and in AA PCa patients by IHC staining of TMA samples. It will be much better if the authors provide the representative IHC images as supplemental figures to directly show UGT2B28 protein expression increasing.
2. According to supplemental figure S2, the UGT2B28 shRNA is not specifically targeting UGT2B28, but also target other members of UGT2B28 family. Have the authors tried multiple shRNAs? Do they have the same effect?
3. The authors demonstrated that UGT2B28 protein expression is regulated by androgen receptor, but the downstream functional phenotype is AR signaling independent. Since the authors have established a series of UGT2B28 KD and OE genetic models, have they performed RNA-Seq to investigate the potential signaling pathways induced by UGT2B28 deficiency or redundancy? It will be much better if the authors can discuss the potential mechanism for the role of UGT2B28 in tumor growth.
4. Please fix the wrong subtitle numbering in P248 and P296.
